# Multi-Cell Testing Topologies for Defect Detection Using Electrochemical Impedance Spectroscopy: A Combinatorial Experiment-Based Analysis

Manuel Ank *, Jonas Göhmann and Markus Lienkamp

Department of Mobility Systems Engineering, Institute of Automotive Technology,
School of Engineering & Design, Technical University of Munich (TUM), 85748 Garching, Germany
* Correspondence: manuel.ank@tum.de; Tel.: +49-89-289-10440

**Abstract:** Given the increasing use of lithium-ion batteries, which is driven in particular by electro-mobility, the characterization of cells in production and application plays a decisive role in quality assurance. The detection of defects particularly motivates the optimization and development of innovative characterization methods, with simultaneous testing of multiple cells in the context of multi-cell setups having been researched to economize on the number of cell test channels required. In this work, an experimental study is presented demonstrating the influence of a defect type in one cell on five remaining interconnected cells in eight combinatorially varied topologies using galvanostatic electrochemical impedance spectroscopy. The results show that regularities related to the interconnection position are revealed when considering the change in the specific resistance $Z_{IM,min}$ at the transition from the charge transfer to the diffusion region between the reference and fault measurements, allowing it to function as a defect identifier in the present scenario. These results and the extensive measurement data provided can serve as a basis for the evaluation and design of multi-cell setups used for simultaneous impedance-based lithium-ion cell characterizations.

**Keywords:** lithium-ion; characterization; electrochemical impedance spectroscopy; multi-cell testing; electric vehicle; defect detection

## 1. Introduction

The importance of lithium-ion cell characterization is increasing as a result of growing worldwide utilization, e.g., in the field of battery electric vehicles (BEVs) [1,2] because quality requirements need to be met, and the surplus value of battery data becomes evident [3–5]. Novel cell quality assessment methodologies and the reliable identification of defects respectively cell anomalies are therefore emerging as a major research area, with simultaneous testing of multiple interconnected cells using electrochemical impedance spectroscopy (EIS) explored in this work [6–8].

### 1.1. Characterization Using Electrochemical Impedance Spectroscopy

Electrochemical impedance spectroscopy (EIS) is a non-destructive technique in which a test object (such as a battery cell) is excited with a generally sinusoidal signal and the system response is measured [9,10]. Either voltage (potentiostatic EIS) or current (galvanostatic EIS) can be used as excitation, with the other being measured as the response signal [11]. By performing EIS measurements at multiple frequencies, a full impedance spectrum can be gained and illustrated in a Nyquist plot, whereby the real part of the impedance is plotted on the abscissa and the negative imaginary part on the ordinate (this visualization prevails due to the strongly capacitive behavior of batteries) [11,12]. Since the frequency-dependent impedance of a cell provides important information about its cell properties, electrochemical impedance spectroscopy is used in battery development and characterization, e.g., to establish battery operating limits, model equivalent

circuits, estimate performance, and monitor state of health (SOH) as well as state of function (SOF) [13,14]. For a detailed overview of the operating principle of the methodology and an aggregate review of the application possibilities in the battery domain, the reader is referred to Mc Carthy et al. [12], Meddings et al. [13], Middlemiss et al. [9], Wildfeuer et al. [11], and Fernández Pulido et al. [15]. The methodology has emerged as a useful tool, with many publications having focused on single-cell-based aging assessment and characterization [10,16–18]. In the interconnected state, Rüther et al. [19] investigated a method for characterizing lithium-ion cells at the multi-cell level using EIS but limited their work to a series interconnection and the detection of aging mechanisms rather than intra-cell variations. Studies of current distributions in parallel-connected cells have furthermore been performed among others by Brand et al. [20] and Bruen et al. [21], although topology variations were not investigated, and no emphasis was placed on EIS. A feasibility study of EIS at the battery pack level was performed by Gong et al. [22], who measured and analyzed pack impedance data from a commercial battery electric vehicle (BEV) via a charging station. However, they did not evaluate the individual cell voltage data and did not focus on defect influences. For a general overview of fault diagnosis methods, the reader is referred to the work of Yu et al. [23]. The authors are not aware of any research investigating a variety of serial and parallel interconnection topologies for the detection of production inhomogeneities in lithium-ion cells using EIS—thus incentivizing this investigation.

### 1.2. Multi-Cell Testing

Multi-cell testing describes the simultaneous testing of interconnected battery cells and builds on the idea of assessing specific individual cell properties in the series-connected or parallel-connected state. In this context, a current $I$ is applied to the main circuit while the voltages of the individual cells $U_1$ to $U_n$ are recorded. This methodology serves as a complement to single-cell-based tests in cell production or application-side quality checks and enables batch-wise processing of the device under test (DUT). Many commonly used characterization methodologies, e.g., resistance determinations (direct current internal resistance (DC-IR) or alternating current internal resistance (AC-IR)) as well as the recording of pseudo open-circuit voltage (pOCV) curves for the calculation of differential voltage and incremental capacity analysis (DVA/ICA), can be implemented in the multi-cell setup, taking into account single-cell scattering and optimally avoiding multiple cycles. Functionality has already been demonstrated at the laboratory level for use in end-of-line testing in cell production [24]. This study builds on these findings by examining the use of EIS in a variety of conceivable interconnection topologies.

### 1.3. Implementation in Quality Control

In the life cycle of a battery cell for vehicle applications, the SOF of the energy storage is determined at various points, whereby the initial characterization in the end-of-line (EOL) test in the cell finishing process at the manufacturer is exceptional because the individual cells can still be scrapped or assigned to another application if necessary [25]. Furthermore, characterizations prior to cell assembly into module and pack in the context of incoming inspections are common for ensuring quality and generating a baseline for application-specific battery lifetime data [26,27]. Finally, characterizations in the interconnected state can again become appropriate during later life cycle phases, e.g., as part of a service check at a dealership or at a remanufacturing facility when evaluating cells for 2nd life reuse or current residual value [28–31]. The non-invasive methodology of multi-cell testing—either with or without characterization via EIS—is in principle applicable to all of the characterization scenarios mentioned because the pristine cells can be interconnected, and an inherent multi-cell system is present after the deployment phase in a module assembly. An implementation via a charging station, as prototypically demonstrated by Gong et al. [22], is also conceivable.

### 1.4. Contributions

In this work, an experimental analysis of eight interconnection topologies of six battery cells is performed using EIS based on the insertion of one cell with deviating performance characteristics (hereinafter referred to as a faulty cell) to enable an evaluation with respect to multi-cell testing applications. The main contributions of this study can be summarized as follows:

- **Combinatorial measurement using galvanostatic EIS**
  All of the topologies investigated were measured in both reference (six intact cells) and fault (insertion of one faulty cell) scenarios using galvanostatic EIS, with the excitation signal applied to the total interconnection and the single-cell voltage responses evaluated.
- **Assessment of position-based influence effects depending on topology**
  Emerging regularities with respect to the position-dependent deviations of the specific resistance $Z_{IM,min}$ of the intact cells in the fault scenarios were identified on the basis of the measured data.
- **Derivation of implications for battery quality testing in multi-cell setups**
  Recommendations for the implementation of multi-cell testing used for the impedance-based quality testing of battery cells were deduced from the findings.

### 1.5. Organization of the Article

The present article is structured as follows: Section 2 presents the topologies and cells under investigation, the measurement setup, the test procedure, as well as the analysis parameter selected. The experimental results of all interconnection topologies are presented and discussed in Section 3, with position-based influence effects and derived recommendations being particularly emphasized. Section 4 concludes by summarizing the findings.

## 2. Experimental

The following section describes the experimental techniques and resources used for this study.

### 2.1. Measurement Setup

A *VSP-3e* multichannel potentiostat with a *SAM-50* stack add-on connected to a *FlexP0060 Booster* (BioLogic SAS, Seyssinet-Pariset, France) was used to perform multi-cell EIS measurements, thus enabling voltage levels of up to 60 V to be investigated. The test procedure control was performed using EC-Lab v11.43 (BioLogic SAS, Seyssinet-Pariset, France) software. The tested devices were secured using individual cell holder model *Battery Holder 30A/60A—Universal Cylindrical cell* (ARBIN Instruments, College Station, TX, USA) and interconnected in parallel or series using connecting cables, depending on the topology to be analyzed. The connecting cables between the cells had a cross-section of $>2.5 \, \text{mm}^2$ and are approved according to IEC 60228 up to a maximum of 20 A at 12 V [32]. Since the EIS was performed with significantly lower currents or power, this cross-section was large enough to prevent safety risks and increased cable resistance. A test chamber model *KB400* (BINDER GmbH, Tuttlingen, Germany) maintained a constant $25 \pm 0.2 \, ^\circ\text{C}$ during all test procedures. The master channel of the booster was connected to the entire cell interconnection setup (cable length 2.5 m), with three further slave channels of the multichannel potentiostat—each with three 1.5 m reference cables ($Ref_1$, $Ref_2$, $Ref_3$) per channel—tapping the voltage of the individual cells. All six individual cell voltages were able to be measured by evaluating $U_{Slave,A} = Ref_1 - Ref_2$ and $U_{Slave,B} = Ref_2 - Ref_3$ per slave channel, as explained in the manufacturer's application note [33]. The experimental setup used to perform the test procedures is demonstrated in Figure A1.

### 2.2. Topologies under Investigation

The present research combinatorially tested series and parallel topologies of six individual cells, thus resulting in four supergroups based on the number of parallel strings: 1p, 2p, 3p, and 6p. The topologies with two parallel strings (2p) and three parallel strings (3p) can each be interconnected in three different configurations: parallel strings, serial modules, or additional cross-connectors. A total of eight different interconnection topologies resulted. A letter and number system is used as nomenclature, where s represents the number of serially connected cells, p the number of parallel connected strings per module, and m the number of series connected modules. Regarding the fault measurements, the first cell at the main positive contact (C1, position 1) was replaced by a cell with deviating characteristics (C1). Figure 1 gives an overview of all interconnection topologies under investigation and their nomenclature.

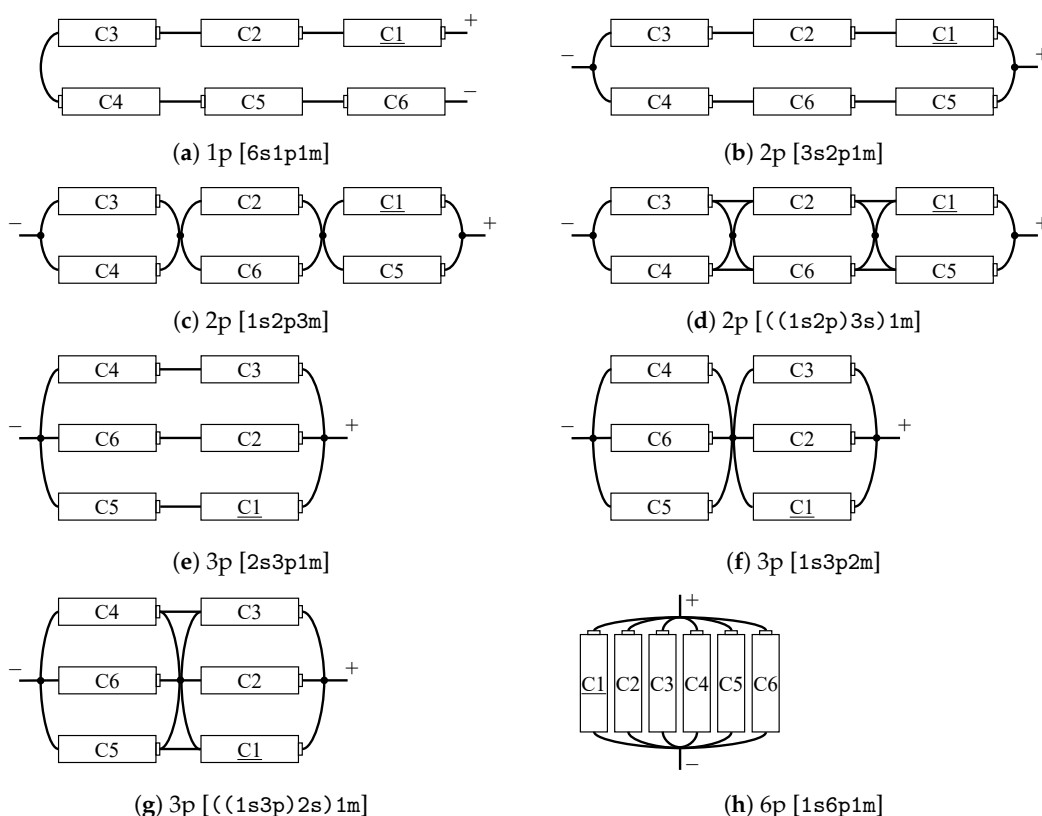

**Figure 1.** Schematic representation of all interconnection topologies investigated. The faulty cell is indicated as C1. An intact cell (C1) was used at position 1 in all of the reference measurements.

### 2.3. Cells under Investigation

The experiments were performed using 18,650 cylindrical cells (manufactured by BAK Battery Co., Ltd., Shenzhen, China) as a representative battery model available to the end user. Six cells with comparable capacitance and internal resistance were selected for this study from a previously characterized cell set (in [27,34]), as well as one outlier cell with respect to these properties introduced during the fault measurements. The selected number of six cells allows the combinatorial investigation of both parallel and serial connected topologies and different configurations for the detection of regularities with reasonable measurement effort and within the device-specific limits (Section 2.1). A summary of the cell specifications can be found in Table 1.

**Table 1.** Specifications of the investigated cell model (obtained from the manufacturer's data sheet) [35].

| Attribute | Specification |
| --- | --- |
| Manufacturer and cell model | BAK N18650CK (flat top) |
| Capacity | 3.05 Ah @ 0.2 C (nominal); 2.95 Ah @ 0.2 C (minimum) |
| Voltage parameters | Nominal voltage: 3.6 V; Charge voltage: 4.2 V; Discharge cut-off voltage: 2.5 V |
| Energy density | 234 $^{Wh}/_{kg}$ |
| Maximum charge current | 1 C ($10\,°C \leq T \leq 45\,°C$); 0.2 C ($10\,°C > T \geq 0\,°C$) |
| Maximum discharge current | 0.5 C ($5\,°C > T \geq -20\,°C$); 1 C ($60\,°C > T \geq 45\,°C$); 2 C ($45\,°C > T \geq 5\,°C$) |
| Battery dimension and mass | Height: $(64.85 \pm 0.25)$ mm; Diameter: $(18.35 \pm 0.15)$ mm; Cell mass: $\leq 47$ g |

In addition, Figure 2 shows the initial characteristics of the six selected cells based on single-cell measurements at $25 \pm 0.2\,°C$ ambient temperature, as obtained using a constant current (CC) C/3 discharge capacity measurement (based on the nominal capacity indicated in Table 1) and potentiostatic EIS measurements at 50% state of charge (SOC) [27,34]. The battery cells used exhibited original production variations and were neither artificially aged nor modified to simulate inhomogeneity.

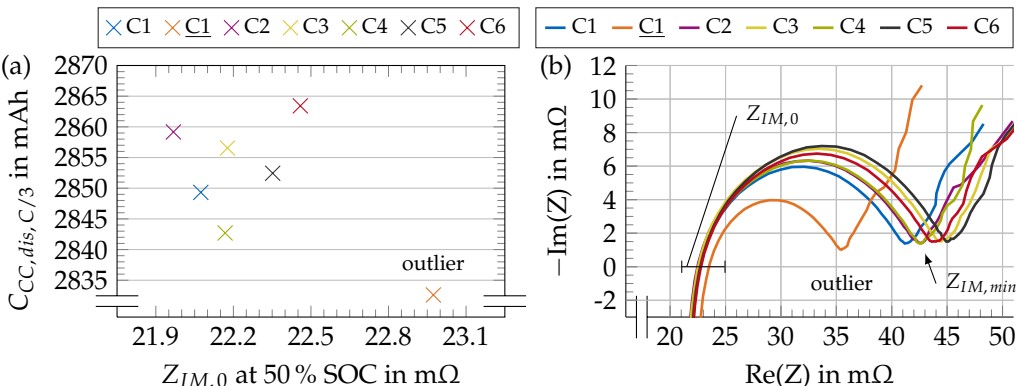

**Figure 2.** Properties of the cells used, as measured by single-cell characterization at $25 \pm 0.2\,°C$ ambient temperature at 50% SOC. (**a**) CC C/3 discharge capacity $C_{CC,dis,C/3}$ and AC-IR $Z_{IM,0}$ at 50% SOC evaluated via potentiostatic EIS measurements (10 kHz to 10 mHz, 10 points per decade (logarithmic spacing), 10 mV excitation amplitude), (**b**) Nyquist plots of these potentiostatic EIS measurements with $Z_{IM,0}$ and $Z_{IM,min}$ indicated [27,34].

### 2.4. Test Procedure

The present individual discharge capacity of all cells was determined before starting the multi-cell test runs (charged to 100% SOC using CC and constant voltage (CV) phases; subsequent CC C/20 discharge until discharge cutoff of 2.5 V). By utilizing the individually measured discharge capacities, all cells were set to 50% SOC based on their individual charge throughput prior to the first multi-cell investigation. Two EIS measurements were performed for each interconnection topology being investigated, whereby a reference measurement was followed by a fault measurement. The reference measurement always refers to the investigation of the respective interconnection topology with six intact cells (C1–C6), while the fault measurement refers to the evaluation with five intact cells and one defective cell in the same interconnection (C1, C2–C6). Based on extensive preliminary investigations, the use of a galvanostatic EIS in the measurement range from 5 kHz to 10 mHz was selected for all of the multi-cell topologies investigated (10 points per decade (logarithmic spacing), 200 mA excitation amplitude per string, 0.1 wait period before each frequency measurement, 4 measures per frequency, activated drift correction). Figure 3 shows the schematic test sequence including the relaxation times between the process steps.

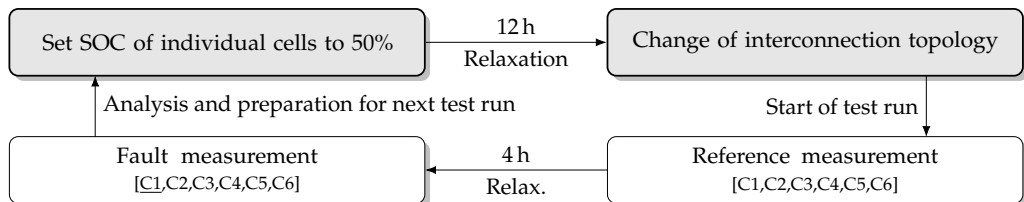

**Figure 3.** Test procedure performed. For the reference measurements, six intact cells each were used, with one defective cell (C1) inserted at cell position 1 for the fault measurements.

### 2.5. Selected Analysis Parameter

The specific resistance $Z_{IM,min}$, which represents the transition from the charge transfer to the diffusion region in the Nyquist plot, is primarily used in the present research to detect an outlier within a cell interconnection due to the impedance behavior of C1. This variable is defined as the diameter of the dominant semicircle in the Nyquist plot, with the local minimum identified at low frequencies and the difference calculated from the ohmic resistance $Z_{IM,0}$ (real part of the impedance at zero crossing in the Nyquist plot) and the abscissa value of this minimum (Figure 2). To enhance visualization of $Z_{IM,min}$ from two DUTs and to compensate for cell connectors increasing the measured ohmic resistance of the individual battery cells, the impedance curves are shifted by their ohmic part in the negative direction on the abscissa to the origin. This is achieved by subtracting the respective abscissa value of $Z_{IM,0}$ from all impedance measurement data of the considered measurement. All plots were additionally smoothed according to a quadratic polynomial using a Savitzky–Golay filter in both axis directions with a window size of eight elements [36]. The calculation of $Z_{IM,min}$ and the post-processing of the data were automated using a MATLAB® (Natick, MA, USA) script (software version R2022b). Additional characteristics, such as those investigated in a serially connected multi-cell setup by Rüther et al. [19], were not used in the present experiment-based analysis but can be analyzed subsequently on the basis of the measurement data obtained.

## 3. Results and Discussion

The experimental results of each interconnection topology are presented in Section 3.1, with particular emphasis and comments on the most important findings. Subsequently, the significance of the deviations is discussed in Section 3.2, and the observed position-based influence effects depending on the topology are highlighted in detail in Section 3.3.

### 3.1. Galvanostatic EIS Measurement Data

Figure 4 presents a comparison between the reference and fault EIS measurements of all multi-cell topologies investigated, i.e., for both the total setup voltage deviations as well as the deviation of the single-cell impedance curves at positions 1 to 6 of the respective topology. Furthermore, all calculated values of $Z_{IM,min}$ of each cell at the topologies investigated are given in Table A1.

In the following, the difference between $Z_{IM,0}$ from the reference and fault measurements of the total setup impedance spectra is denoted as $\Delta Z_0$ deviation, while the difference between $Z_{IM,min}$ from the total setup reference and fault measurements is labeled as $\Delta Z_{min}$ deviation. $\delta Z$ always refers to the deviation of $Z_{IM,min}$ of the referenced individual cells between both measurements. Furthermore, all percentage values in this section are calculated with respect to the respective reference measurement.

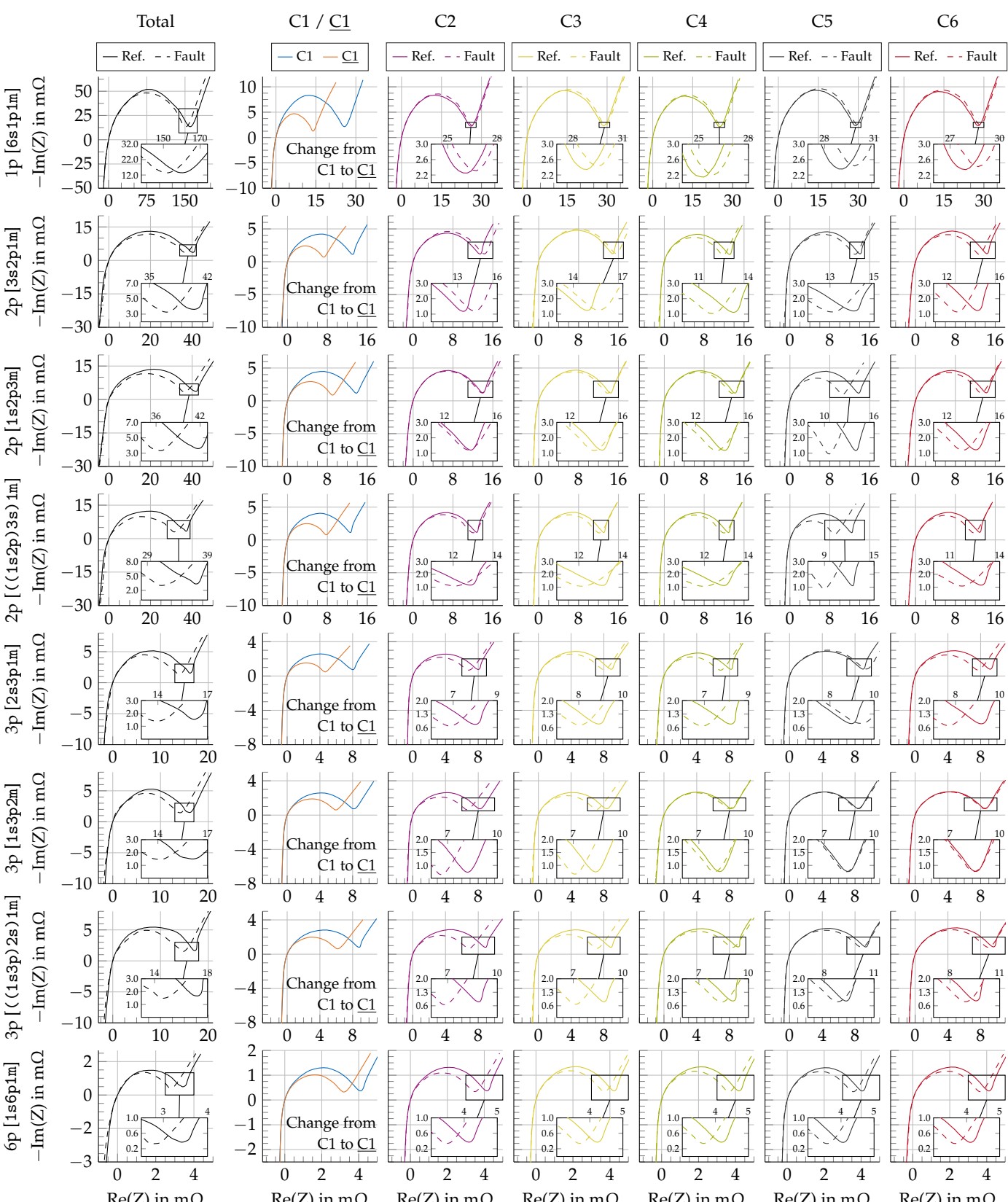

**Figure 4.** Comparison between reference and fault EIS measurements of all multi-cell topologies investigated. The deviation of the total voltage as well as the single-cell impedance curves at positions 1 to 6 of the respective topology are presented. All plots are shifted on the abscissa to $Z_{IM,0} = 0$ to emphasize the variations of $Z_{IM,min}$. Specific values of $Z_{IM,min}$ are given in Table A1.

The `6s1p1m` interconnection exhibited a $\Delta Z_0$ deviation of 3.40 mΩ (1.82%) and a $\Delta Z_{min}$ deviation of 8.90 mΩ (5.54%). Cells C2 to C6 showed a slight $\delta Z$ increase in this topology with an average of +1.73% due to the cell change in position 1 ($\delta Z$ from switching C1 to C1: −46.24%).

The $\Delta Z_0$ deviation of the 3s2p1m interconnection amounts to 0.10 mΩ (0.19%), with a $\Delta Z_{min}$ deviation of 3.36 mΩ (8.32%). For `1s2p3m` and `((1s2p)3s)1m`, values of 0.49 mΩ (0.88%) and 0.07 mΩ (0.13%) as $\Delta Z_0$ deviation as well as 5.14 mΩ (12.33%) and 5.64 mΩ (15.22%) as $\Delta Z_{min}$ deviation resulted when examining the total setup impedance spectra. In the 3s2p1m topology ($\delta Z$ from switching C1 to C1: −44.59%), cells C2 and C3 (same string as C1/C1) showed a $\delta Z$ increase of 5.34%, with cells C4, C5, and C6 (parallel string to C1/C1) exhibiting a $\delta Z$ reduction of −9.29%. When evaluating the 1s2p3m topology ($\delta Z$ from switching C1 to C1: −34.39%), cell C5 (connected in parallel with C1/C1) showed the second largest $\delta Z$ deviation with −23.30%. Cells C2–C4 and C6 exhibit an average $\delta Z$ deviation of −4.12%. The same trend was evident in the `((1s2p)3s)1m` topology ($\delta Z$ from switching C1 to C1: −38.46%) with a $\delta Z$ deviation of C5 of −27.72 and an average $\delta Z$ deviation of cells C2–C4 and C6 of −7.01%.

For the 3p topologies, the values of the $\Delta Z_0$ deviation were 0.14 mΩ (0.47%) (2s3p1m), 0.60 mΩ (1.96%) (1s3p2m), and 0.52 mΩ (1.62%) (((1s3p)2s)1m). The $\Delta Z_{min}$ deviations analogously yielded 2.23 mΩ (13.73%) (2s3p1m), 2.01 mΩ (12.52%) (1s3p2m), and 2.64 mΩ (15.29%) (((1s3p)2s)1m). In the 2s3p1m topology ($\delta Z$ from switching C1 to C1: −42.19%), cell C5 (connected in series with C1/C1) exhibited a $\delta Z$ increase of +4.42%, and the other cells (C2–C6) showed a $\delta Z$ decrease of −14.06%. In the 1s3p2m topology ($\delta Z$ from switching C1 to C1: −26.76%), cells C2 and C3 (directly connected parallel to C1/C1) exhibited a large $\delta Z$ deviation of −19.49% (C2) and −14.69% (C3), while cells C4–C6 revealed an average $\delta Z$ deviation of −2.03%. This trend continued in the ((1s3p)2s)1m topology ($\delta Z$ from switching C1 to C1: −29.50%) with $\delta Z$ deviations of −23.19% for C2 and −18.93% for C3 and an average $\delta Z$ deviation of C4–C6 of −6.50%.

The $\Delta Z_0$ deviation of the 1s6p1m interconnection ($\delta Z$ from switching C1 to C1: −21.82%) amounted to 0.01 mΩ (0.08%), with a $\Delta Z_{min}$ deviation of 0.66 mΩ (19.38%). The cells C2 to C6 (connected in parallel with C1/C1) exhibited an average $\delta Z$ decrease of −12.85%.

Overall, it is evident that the interconnection topologies had a strong influence on the detectability of defective cells. For interconnection topologies with several parallel strings (e.g., `1s6p1m`), it is advisable to consider the total setup voltage measurement data in order to detect the presence of a defect (relative $\Delta Z_{min}$ deviation: 19.38%; averaged $\delta Z$ deviation of all individual cells: 14.52%), whereas, in the case of interconnection topologies with predominantly serially interconnected cells (such as `6s1p1m`), it is advisable to examine the voltage data of the individual cells (relative $\Delta Z_{min}$ deviation: 5.54%; averaged $\delta Z$ deviation of all individual cells: 9.16%). The introduction of additional cross-connectors into the interconnection (((1s2p)3s)1m and ((1s3p)2s)1m) improved detectability in the fault case investigated by increasing the deviations between reference measurement and fault detection.

The influence of cyclic and calendar aging effects on the measurement results was considered to be negligible given an average discharge capacity loss of 1.29% (measurement before and after all tests runs) and a total test time of <3 months. In addition, all of the cells were exposed to the identical conditions, so the intra-cell comparison is unrelated to aging effects.

### 3.2. Assessment of Significant Deviation

To obtain a baseline measurement of the measurement setup scatter present, a nine-fold galvanostatic EIS measurement with invariant test parameters (5 kHz to 10 mHz, 10 points per decade (logarithmic spacing), 600 mA excitation amplitude, wait 0.1 period before each frequency measurement, 4 measures per frequency, activated drift correction, 10 min relaxation between each test run) was performed for a 2s3p1m interconnection topol-

ogy. Using the mean of $\overline{Z}_{IM,0} = 31.17\,\text{m}\Omega$ ($\sigma_{Z_{IM,0}} = 0.16\,\text{m}\Omega$) and $\overline{Z}_{IM,min} = 17.85\,\text{m}\Omega$ ($\sigma_{Z_{IM,min}} = 0.14\,\text{m}\Omega$) and employing a two-sided confidence of $2\sigma$ led to confidence intervals of $CI_{Z_{IM,0}} = [30.84\,\text{m}\Omega; 31.49\,\text{m}\Omega]$ and $CI_{Z_{IM,min}} = [17.56\,\text{m}\Omega; 18.13\,\text{m}\Omega]$. This resulted in a relative deviation of the sigma band of $2\sigma_{Z_{IM,0}} = 1.05\%$ and $2\sigma_{Z_{IM,min}} = 1.59\%$. These limits were used as a decision criterion to distinguish between measurement variation and significant change, i.e., a faulty cell within an interconnection. Additional factors such as fluctuating contact resistances between connectors as well as contact resistance fluctuations due to cell swapping are not taken into account.

Figure 5 illustrates the relative deviations $\Delta Z_0$ and $\Delta Z_{min}$ of the total setup voltage between reference measurement and fault measurement for all interconnection topologies investigated, along with the determined confidence interval. The `6s1p1m` topology exhibited the largest relative $\Delta Z_0$ deviation with 2.04%, while the topologies `1s3p2m` and `((1s3p)2s)1m` also reveal significant relative $\Delta Z_0$ deviations with 1.96% (`1s3p2m`) and 1.62% (`((1s3p)2s)1m`), respectively (Figure 5a). Based on the measurement results, the pure serial connection `6s1p1m` is usable for fault detection via $Z_{IM,0}$, with 2p and 3p topologies only conditionally. The serial modules (`1s2p3m` and `1s3p2m`, respectively) show the largest relative deviation in both supergroups. The assessment of $Z_{IM,0}$ of the pure parallel connection `1s6p1m` does not seem to be suitable for error detection in this fault case.

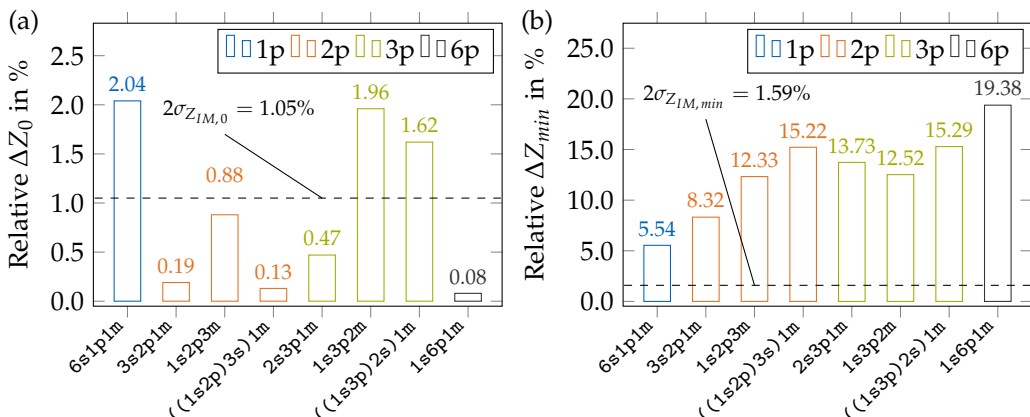

**Figure 5.** Relative deviations of (**a**) the ohmic resistance $Z_{IM,0}$ and (**b**) the specific resistance $Z_{IM,min}$ of the total setup between reference measurement and fault detection for all interconnection topologies investigated ($\Delta Z_0$ and $\Delta Z_{min}$). The number of parallel strings are marked, and the confidence range is indicated.

Examining the deviation of $Z_{IM,min}$ (Figure 5b), all topologies exhibited deviations well above the confidence range and are therefore suitable for successful detection of the exemplary outlier. The relative $\Delta Z_{min}$ deviation was lowest for the `6s1p1m` circuit with 5.54% and highest for the `1s6p1m` circuit with 19.38%. Interconnection variants with cross-connectors (`((1s2p)3s)1m` and `((1s3p)2s)1m`) showed stronger deviations of $Z_{IM,min}$, apart from the relative deviation increasing with an additional number of parallel strands.

Using the example of the investigated outlier cell, use of the specific resistance $Z_{IM,min}$ is recommended for the detection of inhomogeneities based on the measurement data. The absolute deviations are naturally dependent on the specific type of defect present, although the influence of the topology is assumed to be independent. Note that the real part of $Z_{IM,min}$ was rated as less significant for defect identification in the study by Rüther et al. [19], but emerges as the most suitable feature in this study given the fault case depicted. This underlines that depending on the defect scenario, cell type, and the application, alternative features may be significan; thus, the optimal defect detection feature should always be determined iteratively.

### 3.3. Position-Based Influence Effects Depending on Topology

Based on the topologies (Section 2.2) and the measurement data (Section 3.1) examined, two regularities were observed in regard to $Z_{IM,min}$ due to the positioning of the faulty cell at position $\underline{C1}$:

1.  Cells connected directly in parallel to the faulty cell exhibited a deviation of $Z_{IM,min}$, following the trend of the faulty cell
2.  Cells connected directly in series to the faulty cell exhibited a deviation of $Z_{IM,min}$, contrary to the trend of the faulty cell

Figure 6 shows these relationships schematically using four interconnection topologies. It is assumed that these regularities will also be observed in larger topologies with more cells, although additional studies are required for validation.

We hypothesize that the measured effects originate from superimposed phenomena due to the particular topology arrangement, involving marginally unequal voltage drops in series interconnections due to the particular cell characteristics when applying the galvanostatic excitation, and impedance-based partitioning of the excitation current among parallel interconnected cells, resulting in excitation currents partly depending on the respective cell characteristics in the topology. Moreover, with alternating current excitation in the high-frequency range, phenomena such as the skin effect potentially become relevant, complicating a simulative examination of the observed effects [37–39]. Additional influences due to the measurement setup (ohmic resistances due to slightly deviating contact resistances at connectors as well as inductances due to the connection layout) can also not be completely excluded despite utmost care in the design and execution of the experiments.

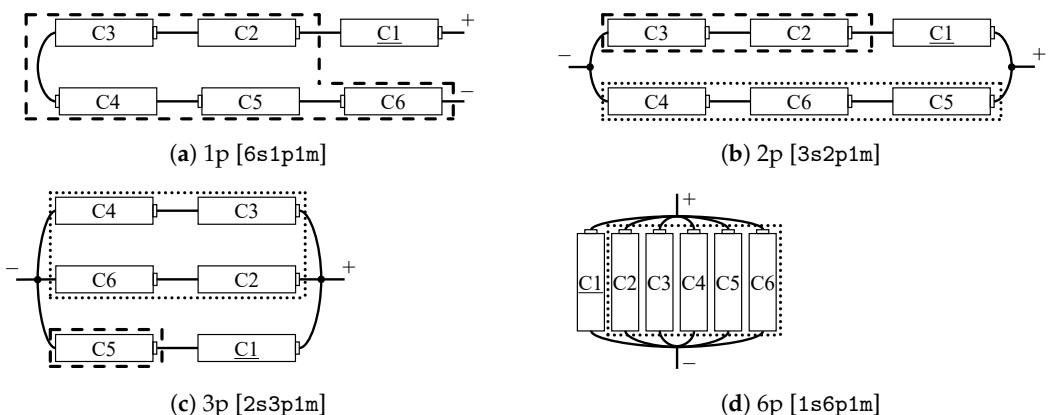

**(a)** 1p [6s1p1m]      **(b)** 2p [3s2p1m]

**(c)** 3p [2s3p1m]      **(d)** 6p [1s6p1m]

**Figure 6.** Schematic representation of observable regularities using four interconnection topologies as an example. The faulty cell is indicated as $\underline{C1}$. Deviations according to regularity (1) are framed with dotted lines, and deviations according to regularity (2) are framed with dashed lines.

### 3.4. Recommendations for Implementation and Outlook

Based on the experimental data, the following recommendations can be derived for the implementation of defect identification using EIS in a multi-cell setup: First, the measured impedance spectra of the single cells cannot be regarded as identical to a single-cell-based characterization, since inter-cell influences arise, and—depending on the position of the individual voltage taps—ohmic components due to additional cable resistances must be taken into account. Second, it is necessary to both know and to take into account the regularities which occur in a manner depending on topology (Section 3.3) in order to correctly classify conclusions about the actual defective cell. In addition, it seems necessary and appropriate to conduct extensive preliminary studies with intact DUTs in order to obtain reference measurement data and to calibrate any clustering procedure used for defect classification. In general, depending on the application, consideration must be given to whether a complex interconnection topology is advantageous for usage in characterization

tests over single-cell tests. As a default application-oriented multi-cell setup, a pure series interconnection is recommended to limit complexity.

Future studies might investigate variation in the position of the defective cell, reverse calculation to the localization of the defective cell, the presence of multiple overlapping defective cells, simulative modeling of the observed effects, scaling of the number of cells, and divergent defect characteristics up to complete cell failure along with the effect in the interconnection. Moreover, an evaluation using additional characteristics might be based on the measurement data provided, and a corresponding clustering procedure could be developed.

## 4. Conclusions

Given the initial and lifetime variations in cell properties, the characterization of lithium-ion cells at several points during the life cycle is useful, especially for applications with high demands such as deployment in BEVs. The present paper describes experimental research on the effect of one battery with deviating impedance characteristics on interconnected battery cells in a multi-cell setup based on eight interconnection topologies using non-destructive EIS measurements. The main results of the study can be concluded as follows:

- The difference in the specific resistance $Z_{IM,min}$ induced by the faulty cell is dependent on the interconnection topology and is usable as a defect indicator in the present scenario.
- Two regularities are observed with respect to the topologies, with cells connected directly in parallel to the faulty cell exhibiting a deviation of the impedance characteristic following the trend of the faulty cell and cells connected directly in series opposing this trend.
- To limit complexity, a pure series interconnection is recommended as a default application-oriented multi-cell setup.

These findings can serve as a basis for evaluating and designing multi-cell setups, which can be used for simultaneous impedance-based characterization of multiple batteries. Future studies might both elaborate upon various aspects of the experimental setup (e.g., investigations with larger interconnection topologies) and perform advanced analyses with additional characteristics based on the measured data.

**Supplementary Materials:** The following supporting information can be downloaded at: https://www.mdpi.com/article/10.3390/batteries9080415/s1.

**Author Contributions:** Conceptualization, M.A.; methodology, M.A. and J.G.; software, M.A. and J.G.; validation, M.A., J.G. and M.L.; formal analysis, M.A. and J.G.; investigation, M.A.; resources, M.L.; data curation, M.A. and J.G.; writing—original draft preparation, M.A. and J.G.; writing—review and editing, M.A., J.G. and M.L.; visualization, M.A.; supervision, M.L.; project administration, M.A. and M.L.; funding acquisition, M.L. All authors have read and agreed to the published version of the manuscript.

**Funding:** This research was funded by the German Federal Ministry of Education and Research (BMBF) within the project "OptiPro" under grant number 03XP0364B as part of the competence cluster "InZePro".

**Institutional Review Board Statement:** Not applicable.

**Informed Consent Statement:** Not applicable.

**Data Availability Statement:** We want to give any researcher access to our results without any limits. The raw measurement data of all experiments are therefore provided in the supplementary material as open source alongside the article.

**Acknowledgments:** We would like to thank Florian Biechl and the staff of the electric lab of the Institute of Automotive Technology for their support during the measurements.

**Conflicts of Interest:** The authors declare that they have no known competing financial interests or personal relationships that could have appeared to influence the work reported in this paper.

## Abbreviations

The following abbreviations are used in this manuscript:

| | |
|---|---|
| AC-IR | alternating current internal resistance |
| BEV | battery electric vehicle |
| CC | constant current |
| CV | constant voltage |
| DC-IR | direct current internal resistance |
| DUT | device under test |
| DVA | differential voltage analysis |
| EIS | electrochemical impedance spectroscopy |
| EOL | end-of-line |
| ICA | incremental capacity analysis |
| pOCV | pseudo open-circuit voltage |
| SOC | state of charge |
| SOF | state of function |
| SOH | state of health |

## Appendix A. Measurement Results and Experimental Setup

The computed values of $Z_{IM,min}$ of each cell and the total setup at every topology analyzed are presented in Table A1. The experimental setup used to perform the test procedures is demonstrated in Figure A1.

**Table A1.** Calculated values of the real part of $Z_{IM,min}$ of the total setup and each cell at every topology investigated. All measurements performed as a galvanostatic EIS, as described in Section 2.4. All values in mΩ.

| Topology | | Measurement Type | Total | C1 / C1 | C2 | C3 | C4 | C5 | C6 |
|---|---|---|---|---|---|---|---|---|---|
| 1p | 6s1p1m | Reference | 160.60 | 25.82 | 26.18 | 28.84 | 25.31 | 29.11 | 27.95 |
| | | Fault | 151.70 | 13.88 | 26.75 | 29.52 | 25.93 | 29.83 | 28.70 |
| 2p | 3s2p1m | Reference | 40.38 | 13.10 | 13.45 | 14.85 | 13.13 | 14.09 | 14.34 |
| | | Fault | 37.02 | 7.26 | 14.29 | 15.51 | 11.85 | 13.07 | 12.78 |
| 2p | 1s2p3m | Reference | 41.69 | 13.83 | 14.01 | 14.44 | 14.21 | 13.82 | 14.21 |
| | | Fault | 36.55 | 9.07 | 13.88 | 13.91 | 13.56 | 10.60 | 13.17 |
| 2p | ((1s2p)3s)1m | Reference | 37.05 | 12.65 | 12.79 | 13.10 | 12.91 | 12.43 | 12.91 |
| | | Fault | 31.41 | 7.79 | 12.15 | 12.13 | 12.08 | 8.98 | 11.72 |
| 3p | 2s3p1m | Reference | 16.24 | 8.01 | 8.00 | 8.93 | 8.18 | 8.99 | 9.18 |
| | | Fault | 14.01 | 4.63 | 6.85 | 8.06 | 6.89 | 9.38 | 7.70 |
| 3p | 1s3p2m | Reference | 16.05 | 8.16 | 8.21 | 8.34 | 8.24 | 8.48 | 8.54 |
| | | Fault | 14.04 | 5.97 | 6.61 | 7.12 | 7.97 | 8.32 | 8.46 |
| 3p | ((1s3p)2s)1m | Reference | 17.27 | 8.77 | 8.87 | 8.86 | 9.02 | 9.28 | 9.60 |
| | | Fault | 14.63 | 6.18 | 6.81 | 7.18 | 8.37 | 8.79 | 8.93 |
| 6p | 1s6p1m | Reference | 3.41 | 4.08 | 4.13 | 4.25 | 4.18 | 4.05 | 4.11 |
| | | Fault | 2.75 | 3.19 | 3.48 | 3.69 | 3.64 | 3.56 | 3.64 |

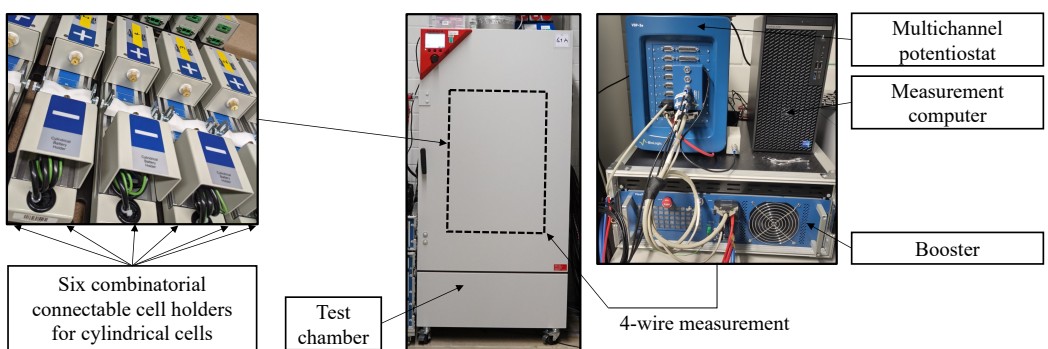

**Figure A1.** Experimental setup used to perform the test procedures. For details refer to Section 2.1.

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
