# Peer review of "Multi-Cell Testing Topologies for Defect Detection Using Electrochemical Impedance Spectroscopy: A Combinatorial Experiment-Based Analysis"

_batteries, doi:10.3390/batteries9080415_

Round 1

Reviewer 1 Report

This paper explores the importance of characterizing lithium-ion batteries during production and application, particularly in the context of electromobility. It investigates the impact of a defect in one battery on five interconnected batteries using galvanostatic electrochemical impedance spectroscopy. The study reveals regularities related to the interconnection position, allowing the change in charge transfer resistance to serve as a defect identifier. The results and extensive measurement data provide a basis for evaluating and designing multi-cell setups for simultaneous impedance-based lithium-ion cell characterizations. And there are some limitations and shortcomings in the research that need further improvement.

1. In the abstract, it is advisable to quantitatively illustrate the superiority of the proposed multi-cell testing topology discussed in this paper.

2. The introduction initially references 12 articles. To improve the quality of references, the author should carefully select core literature relevant to the research topic.

3. To enhance clarity for readers, it is advisable to include a table in the introduction section, highlighting the limitations of previous studies and underscoring the significance of this research.

4. Consider adding an experimental setup physical diagram, to provide a clearer understanding of the research apparatus.

5. Ensure to provide clear explanations for the calculation formulas of relevant research parameters.

6. Please include details about the specific version of software used for data processing, along with a comprehensive explanation of the calculation process.

7. In the discussion section, it would be valuable to incorporate comparative analyses with relevant literature to further support the theoretical insights presented.

8. Provide a rationale for selecting six cells as the basis for the multi-cell topology studied in this paper. Additionally, explore whether extending this study to a larger number of cells yields similar conclusions.

9. It is suggested to thoroughly investigate the underlying reasons for the variations in relative deviations among different topological configurations in the discussion section, rather than merely presenting the results.

10. Revise the conclusion section, highlighting the unique contributions of this paper in a concise and organized manner, possibly using bullet points.

The writing is acceptable.

Reviewer 2 Report

1. It would be beneficial to provide a comprehensive overview of the existing defect detection techniques.

2. Please discuss the potential implications of the findings of this work on real-world applications and how this method could be practically implemented.

3. Will the measurement data in this work be published together with this paper?

4. The main contributions should be more clear in the abstract and conclusion.

Reviewer 3 Report

The authors analyze the implications of the position of a fault battery in a multi-cell testing by applying galvanostatic impedance spectroscopy. This work is a technical approach that can potentially help in developing a serial test method for controlling the production quality of batteries. The paper is well written and structured. I would recommend ist publication providing that some minor changes are made to improve its readability as detailed below. 

1.       Subsection 2.1, lines 105-112: it is not very clear, how do you proceed to evaluate the individual cell voltages. Perhaps a small scheme would be helpful for readers not very familiarized with this type of techniques.

2.       There are a lot of acronyms throughout the paper. What about a list of them at the end of the article with an explanation?

3.       Line 132: what is CC C/3?

4.       Line 140: does implies a discharge cutoff of 2.5 V a 50% SOC? Please, explain if that is the same?

5.       I think that to indicate a charge transfer resistance with f ZIM is not namely fortunate, because it is usually used for “frequency”. Furthermore, the minimum of the imaginary part in the Nyquist diagram does not necessarily matches with the extrapolation of the first loop to the real axis! Please comment!

6.       Line 258-259: I don’t think that the variation of the intensity of the sinusoidal current input should be a reason for the different diagrams. The linearity of the system is sine qua non for the impedance measurement. A dependency of the intensity of the input signal implies a non-linearity and this should invalidate your results. Please reconsider your explanation here.    

Reviewer 4 Report

The authors reported “multi-cell testing topologies for defect detection using electrochemical impedance spectroscopy: A combinatorial experiment-based analysis” that highlights the evaluation and design of impedance-based lithium-ion cells. This study presents a systematic methodology on electrochemical impedance spectroscopy for detection of the charge transfer resistance. The manuscript is clearer and well structured; and some typos are required before publishing.

The manuscript is clearer and well structured; and some typos are required before publishing.

Round 2

Reviewer 1 Report

The authors improved the paper quality based on the reviewer's comments.

The paper can be accepeted as it is.

Reviewer 2 Report

The revision has covered my comments. 

Reviewer 3 Report

The Authors have revised the manuscript satisfactorily. Thus, I recommend now the publishing of this article.